# Paris-CARLA-3D: A Real and Synthetic Outdoor Point Cloud Dataset for Challenging Tasks in 3D Mapping

**Jean-Emmanuel Deschaud [1,]*, David Duque [2], Jean Pierre Richa [1], Santiago Velasco-Forero [2],
Beatriz Marcotegui [2] and François Goulette [1]**

1   MINES ParisTech, PSL University, Centre for Robotics, 75006 Paris, France;
    jean-pierre.richa@mines-paristech.fr (J.P.R.); francois.goulette@mines-paristech.fr (F.G.)
2   MINES ParisTech, PSL University, Centre for Mathematical Morphology, 77300 Fontainebleau, France;
    david.duque@mines-paristech.fr (D.D.); santiago.velasco@mines-paristech.fr (S.V.-F.);
    beatriz.marcotegui@mines-paristech.fr (B.M.)
*   Correspondence: jean-emmanuel.deschaud@mines-paristech.fr

**Abstract:** Paris-CARLA-3D is a dataset of several dense colored point clouds of outdoor environments built by a mobile LiDAR and camera system. The data are composed of two sets with synthetic data from the open source CARLA simulator (700 million points) and real data acquired in the city of Paris (60 million points), hence the name Paris-CARLA-3D. One of the advantages of this dataset is to have simulated the same LiDAR and camera platform in the open source CARLA simulator as the one used to produce the real data. In addition, manual annotation of the classes using the semantic tags of CARLA was performed on the real data, allowing the testing of transfer methods from the synthetic to the real data. The objective of this dataset is to provide a challenging dataset to evaluate and improve methods on difficult vision tasks for the 3D mapping of outdoor environments: semantic segmentation, instance segmentation, and scene completion. For each task, we describe the evaluation protocol as well as the experiments carried out to establish a baseline.

**Keywords:** dataset; LiDAR; mobile mapping; laser scanning; 3D mapping; synthetic; point cloud; outdoor; semantic; scene completion

## 1. Introduction

Data in the form of a 3D point cloud are becoming increasingly popular. There are mainly three families of 3D data acquisition: photogrammetry (Structure from Motion and Multi-View Stereo from photos), RGB-D or structured light scanners (for small objects or indoor scenes), and static or mobile LiDARs (for outdoor scenes). The advantage of this last family (mobile LiDARs) is their ability to acquire large volumes of data. This results in many potential applications: city mapping, road infrastructure management, construction of HD maps for autonomous vehicles, etc.

There are already many datasets published on the first two families, but few are available on outdoor mapping. However, there are still many challenges in the ability to analyze outdoor environments from mobile LiDARs. Indeed, the data contain a lot of noise (due to the sensor but also to the mobile system) and have significant local anisotropy and also missing parts (due to occlusion of objects).

The main contributions of this article are as follows:

- the publication of a new dataset, called Paris-CARLA-3D (PC3D in short)—synthetic and real point clouds of outdoor environments; the dataset is available at the following URL: https://npm3d.fr/paris-carla-3d, accessed on 15 October 2021;
- the protocol and experiments with baselines on three tasks (semantic segmentation, instance segmentation, and scene completion) based on this dataset.

## 2. Related Datasets

With the democratization of 3D sensors, there are more and more point cloud datasets available. We can see in Table 1 a list of datasets based on 3D point clouds. We have listed only datasets available in the form of a point cloud. We have, therefore, not listed the datasets such as NYUv2 [1], which do not contain the poses (trajectory of the RGB-D sensor) and thus do not allow for producing a dense point cloud of the environment. We are also only interested in terrestrial datasets, which is why we have not listed aerial datasets such as DALES [2], Campus3D [3] or SensatUrban [4].

First, in Table 1, we performed a separation according to the environment: the indoor datasets (mainly from RGB-D sensors) and the outdoor datasets (mainly from LiDAR sensors). For outdoor datasets, we also made the distinction between perception datasets (to improve perception tasks for the autonomous vehicle) and mapping datasets (to improve the mapping of the environment). For example, the well-known SemanticKITTI [5] consists of a set of LiDAR scans from which it is possible to produce a dense point cloud of the environment with the poses provided by SLAM or GPS/IMU, but the associated tasks (such as semantic segmentation or scene completion) are only centered on a LiDAR scan for the perception of the vehicle. This is very different from the dense point clouds of mapping systems such as Toronto-3D [6] or our Paris-CARLA-3D dataset. For the semantic segmentation and scene completion tasks, SemanticKITTI [7] uses only one single LiDAR scan as input (one rotation of the LiDAR). In our dataset, we wish to find the semantic and seek to complete the "holes" on the dense point cloud after the accumulation of all LiDAR scans.

Table 1 thus shows that Paris-CARLA-3D is the only dataset to offer annotations and protocols that allow for working on semantic, instance, and scene completion tasks on dense point clouds for outdoor mapping.

**Table 1.** Point cloud datasets for semantic segmentation (SS), instance segmentation (IS), and scene completion (SC) tasks. RGB means color available on all points of the point clouds. In parentheses for SS, we show only the number of classes evaluated (the annotation can have more classes).

| Scene | Type | Dataset (Year) | World | # Points | RGB | Tasks | | |
|---|---|---|---|---|---|---|---|---|
| | | | | | | SS | IS | SC |
| Indoor | Mapping | SUN3D [8] (2013) | Real | 8 M | Yes | ✓ (11) | ✓ | |
| | | SceneNet [9] (2015) | Synthetic | - | Yes | ✓ (11) | ✓ | ✓ |
| | | S3DIS [10] (2016) | Real | 696 M | Yes | ✓ (13) | ✓ | ✓ |
| | | ScanNet [11] (2017) | Real | 5581 M | Yes | ✓ (11) | ✓ | ✓ |
| | | Matterport3D [12] (2017) | Real | 24 M | Yes | ✓ (11) | ✓ | ✓ |
| Outdoor | Perception | PreSIL [13] (2019) | Synthetic | 3135 M | Yes | ✓ (12) | ✓ | |
| | | SemanticKITTI [5] (2019) | Real | 4549 M | No | ✓ (25) | ✓ | ✓ |
| | | nuScenes-Lidarseg [14] (2019) | Real | 1400 M | Yes | ✓ (32) | ✓ | |
| | | A2D2 [15] (2020) | Real | 1238 M | Yes | ✓ (38) | ✓ | |
| | | SemanticPOSS [16] (2020) | Real | 216 M | No | ✓ (14) | ✓ | |
| | | SynLiDAR [17] (2021) | Synthetic | 19,482 M | No | ✓ (32) | | |
| | | KITTI-CARLA [18] (2021) | Synthetic | 4500 M | Yes | ✓ (23) | ✓ | |
| | Mapping | Oakland [19] (2009) | Real | 2 M | No | ✓ (5) | | |
| | | Paris-rue-Madame [20] (2014) | Real | 20 M | No | ✓ (17) | ✓ | |
| | | iQmulus [21] (2015) | Real | 12 M | No | ✓ (8) | ✓ | |
| | | Semantic3D [7] (2017) | Real | 4009 M | Yes | ✓ (8) | | |
| | | Paris-Lille-3D [22] (2018) | Real | 143 M | No | ✓ (9) | ✓ | |
| | | SynthCity [23] (2019) | Synthetic | 368 M | Yes | ✓ (9) | | |
| | | Toronto-3D [6] (2020) | Real | 78 M | Yes | ✓ (8) | | |
| | | TUM-MLS-2016 [24] (2020) | Real | 41 M | No | ✓ (8) | | |
| | | **Paris-CARLA-3D (2021)** | **Synthetic+Real** | **700 + 60 M** | **Yes** | ✓ (23) | ✓ | ✓ |

## 3. Dataset Construction

This dataset is divided into two parts: a first set of real point clouds (60 M points) produced by a LiDAR and camera mobile system, and a second synthetic set produced by the open source CARLA simulator. Images of the different point clouds and annotations are available in Appendix B.

### 3.1. Paris (Real Data)

To create the Paris-CARLA-3D (PC3D) dataset, we developed a prototype mobile mapping system equipped with a LiDAR (Velodyne HDL32) tilted at 45° to the horizon and a 360° poly-dioptric camera Ladybug5 (composed of 6 cameras). Figure 1 shows the rear of the vehicle with the platform containing the various sensors.

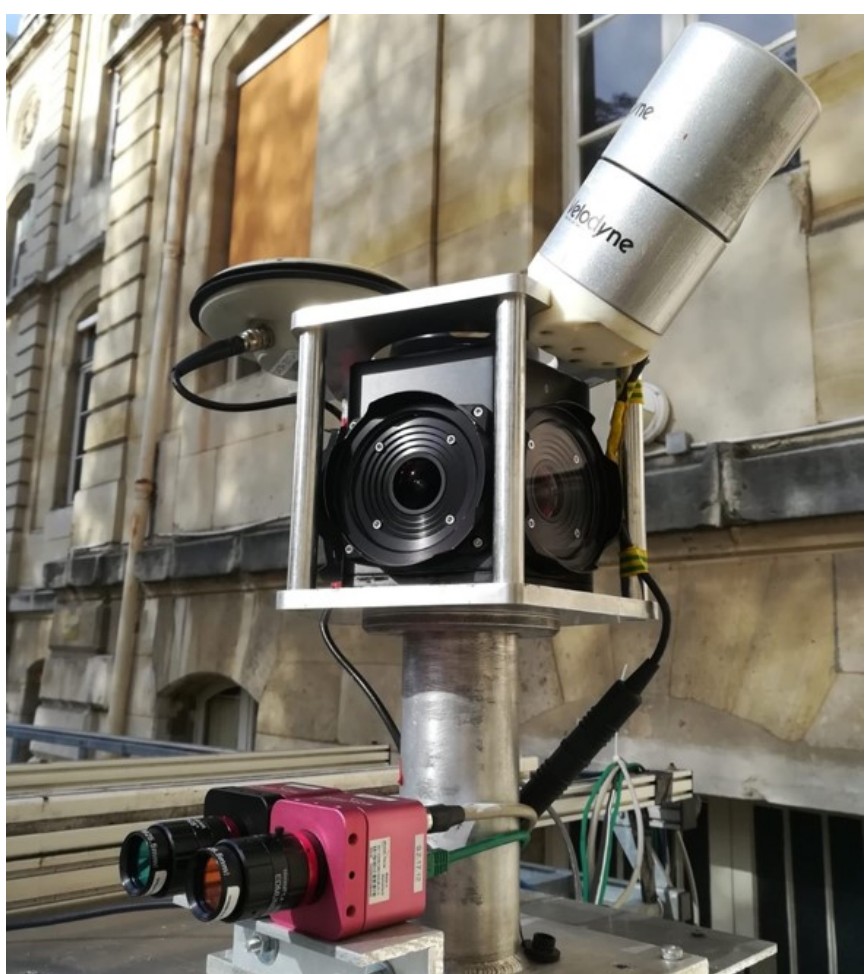

**Figure 1.** Prototype acquisition system used to create the PC3D dataset in the city of Paris. Sensors: Velodyne HDL32 LiDAR, Ladybug5 360° camera, Photonfocus MV1 16-band VIR and 25-band NIR hyperspectral cameras (hyperspectral data are not available in this dataset; they cannot be used in mobile mapping due to the limited exposure time).

The acquisition was made on a part of Saint-Michel Avenue and Soufflot Street in Paris (a very dense urban area with many static and dynamic objects, presenting challenges for 3D scene understanding).

Unlike autonomous vehicle platforms such as KITTI [25] or nuScenes [14], the LiDAR is positioned at the rear and is tilted to allow scanning of the entire environment, thus allowing the buildings and the roads to be fully mapped.

To create the dense point clouds, we aggregated the LiDAR scans using a precise SLAM LiDAR based on IMLS-SLAM [26]. IMLS-SLAM only uses LiDAR data for the construction

of the dataset. However, our platform is equipped with a high-precision IMU (LANDINS iXblue) and a GPS RTK. However, in a very dense environment (with tall buildings), an IMU + GPS-based localization (even with post-processing) achieves lower performance than a good LiDAR odometry (thanks to the buildings). The important hyperparameters of IMLS-SLAM used for Paris-CARLA are: $n = 30$ scans, $s = 600$ keypoints/scan, $r = 0.50$ m for neighbor search (explanations of these parameters are given in [26]). The drift of the IMLS-SLAM odometry is less than 0.40 % with no failure case (failure = no convergence of the algorithm). The quality of the odometry makes it possible to consider this localization as "ground truth".

The 360° camera was synchronized and calibrated with the LiDAR. The 3D data were colored by projecting on the image (with a timestamp as close as possible to the LiDAR timestamp) each 3D point of the LiDAR.

The final data were split according to the timestamp of points in six files (in binary ply format) with 10 M points in each file. Each point has many attributes stored: *x, y, z, x_lidar_position, y_lidar_position, z_lidar_position, intensity, timestamp, scan_index, scan_angle, vertical_laser_angle, laser_index, red, green, blue, semantic, instance*.

For the data annotation, this was done entirely manually with 3 people involved in 3 phases. In phase 1, the dataset was divided into two parts, with one person annotating each part (approximately 100 h of labeling per person). In phase 2, a verification of the annotations was performed by the other person on the part that he did not annotate with feedback and corrections. In phase 3, a third person outside the annotation carried out the verification of the labels on the entire dataset and a consistency check with the annotation in CARLA. The software used for annotation and checks was CloudCompare. The total time in human effort was approximately 300 h to obtain very high quality, as visible in Figure 2. The annotation of the data consisted of adding the semantic information (23 classes) and instance information for the *vehicle* class. The classes are the same as those defined in the CARLA simulator, making it possible to test transfer methods from synthetic to real data.

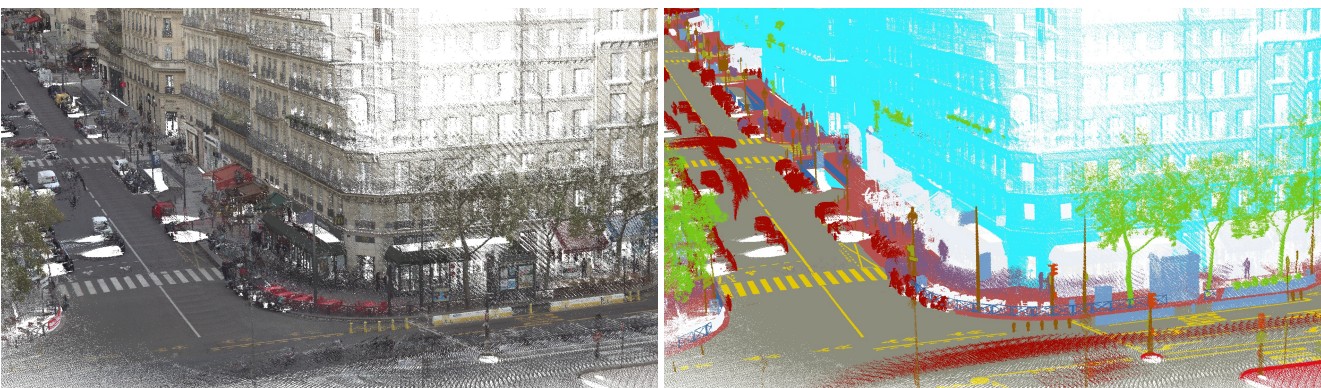

**Figure 2.** Paris-CARLA-3D dataset: (**left**) Paris point clouds with color information on LiDAR points; (**right**) manual semantic annotation of the LiDAR points (using the same tags from the CARLA simulator). We can see the large number of details in the manual annotation.

### 3.2. CARLA (Synthetic Data)

The open source CARLA simulator [27] allows for the simulation of the LiDAR and camera sensors in virtual outdoor environments. Starting from our mobile system (with Velodyne HDL32 and Ladybug5 360° camera), we created a virtual vehicle with the same sensors positioned in the same way as on our real platform. We then launched simulations to generate point clouds in the seven maps of CARLA v0.9.10 (called "Town01" to "Town07"). We finally assembled the scans using the ground truth trajectory and then kept one point cloud with 100 million points per town.

The 3D data were colored by projecting on the image (with a timestamp as close as possible to the LiDAR timestamp) each 3D point of the LiDAR. We used the same colorization process used with the real data from Paris.

The final data were stored in seven files (in binary ply format) with 100 M points in each file (one file = one town = one map in CARLA). We kept the following attributes per point: *x*, *y*, *z*, *x_lidar_position*, *y_lidar_position*, *z_lidar_position*, *timestamp*, *scan_index*, *cos_angle_lidar_surface*, *red*, *green*, *blue*, *semantic*, *instance*, *semantic_image*.

The annotation of CARLA data was automatic, thanks to the simulator with semantic information (23 classes) and instances (for the *vehicle* and *pedestrian* classes). We also kept during the colorization process the semantic information available in images in the attribute *semantic_image*.

### 3.3. Interest in Having Both Synthetic and Real Data

One of the interests of the Paris-CARLA-3D dataset is to have both synthetic and real data. The synthetic data are built with a virtual platform as close as possible to the real platform, allowing us to reproduce certain classic acquisition system issues (such as the difference in point of view of LiDAR and cameras sensors, creating color artifacts on the point cloud). Synthetic data are relatively easy to produce in large quantities (here 700 M points) and with ground truth without additional work for various 3D vision tasks such as classes or instances. It is thus of increasing interest to develop new methods on synthetic data but there is no evidence that they work on real data. With Paris-CARLA-3D and therefore with particular attention to having the same annotations between synthetic and real data, a method can be learned on synthetic data and tested on real data (which we will do in Section 5.2.6). However, we will see that the results remain limited. An interesting and promising approach will be to learn on synthetic data and to develop methods of performing unsupervised adaptation on real data. In this way, the methods will be able to learn from the large amount of data available in synthetic and, even better, from classes or objects that do not frequently meet in reality.

## 4. Dataset Properties

Paris-CARLA-3D has a linear distance of 550 m in Paris and approximately 5.8 km in CARLA (the same order of magnitude as the number of points ($\times 10$) between synthetic and real). For the real part, this represents three streets in the center of Paris. The area coverage is not large but the number and variety of urban objects, pedestrian movements, and vehicles is important: it is precisely this type of dense urban environment that is challenging to analyze.

### 4.1. Statistics of Classes

Paris-CARLA-3D is split into seven point clouds for the synthetic CARLA data, Town1 ($T_1$) to Town7 ($T_7$), and six point clouds for the real data of Paris, Soufflot0 ($S_0$) to Soufflot5 ($S_5$).

For CARLA data, cities can be divided into two groups: urban ($T_1$, $T_2$, $T_3$, and $T_5$) and rural ($T_4$, $T_6$, $T_7$).

For the Paris data, the point clouds can be divided into two groups: those near the Luxembourg Garden with vegetation and wide roads ($S_0$ and $S_1$), and those in a more dense urban configuration with buildings on both sides ($S_2$, $S_3$, $S_4$ and $S_5$).

The detailed distribution of the classes is presented in Appendix A.

### 4.2. Color

The point clouds are all colored (RGB information per point coming from cameras synchronized with the LiDAR), making it possible to test methods using geometric and/or appearance modalities.

*4.3. Split for Training*

For the different tasks presented in this article, according to the distribution of the classes, we chose to split the dataset into the following Train/Val/Test sets:

- Training data: $S_1$, $S_2$ (Paris); $T_2$, $T_3$, $T_4$, $T_5$ (CARLA);
- Validation data: $S_4$, $S_5$ (Paris); $T_6$ (CARLA);
- Test data: $S_0$, $S_3$ (Paris); $T_1$, $T_7$ (CARLA).

*4.4. Transfer Learning*

Paris-CARLA-3D is the first mapping dataset that is based on both synthetic and real data (with the same "platform" and the same data annotation). Indeed, simulators are becoming more and more reliable, and the fact of being able to transfer a method from a synthetic dataset created by a simulator to a real dataset is a line of research that could be important in the future.

We will now describe three 3D vision tasks using this new Paris-CARLA-3D dataset.

**5. Semantic Segmentation (SS) Task**

Semantic segmentation of point clouds is a task of increasing interest over the last several years [4,28]. This is an important step in the analysis of dense data from mobile LiDAR mapping systems. In Paris-CARLA-3D, the points are annotated point-wise with 23 classes whose tags are those defined in the CARLA simulator [27]. Figure 2 shows an example of semantic annotation in the Paris data.

*5.1. Task Protocol*

We introduce the task protocol to perform semantic segmentation in our dataset, allowing future work to build on the initial results presented here. We have many different objects belonging to the same class, as it the case in towns in the real world. This increases the complexity of the semantic segmentation task.

The evaluation of the performance in semantic segmentation tasks relies on True Positives (TP), False Positives (FP), True Negatives (TN), and False Negatives (FN) for each class $c$. These values are used to calculate the following metrics by class $c$: precision $P_c$, recall $R_c$, and Intersection over Union $IoU_c$. To describe the performance of methods, we usually report mean IoU as $mIoU$ Equation (1) and Overall Accuracy as $OA$.

$$mIoU = \frac{1}{C} \sum_{c=1}^{C} \frac{TP_c}{TP_c + FN_c + FP_c} \tag{1}$$

where $C$ is the number of classes.

*5.2. Experiments: Setting a Baseline*

In this section, we present experiments performed under different configurations in order to demonstrate the relevance and high complexity of PC3D. We provide two baselines for all experiments with PointNet++ [29] and KPConv [30] architectures, two models widely used in semantic segmentation and which have demonstrated good performance on different datasets [4]. A recent survey with a detailed explanation of the different approaches to performing semantic segmentation on point clouds from urban scenes can be found at [28,31].

One of the challenges of dense outdoor point clouds is that they cannot be kept in memory, due to the high number of points. In both baselines, we used a subsampling strategy based on sphere selection. The spheres were selected using a weighted random, with the class rates of the dataset as probability distributions. This technique permits us to choose spheres centered in less populated classes. We evaluated different radius spheres ($r = 2$, 5, 10, 20 m) and we evidenced that using $r = 10$ m was a good compromise between computational cost and performance. On the one hand, small spheres are fast to process but provide poor information about the environment. On the other hand, large spheres provide richer information about the environment but are too expensive to process.

5.2.1. Baseline Parameters

The first baseline is based on the Pointnet++ architecture, commonly used in deep learning applications. We selected the architecture provided by the authors [29]. It is composed of three abstraction layers as the feature extractor and three MLP as the last part of the model. The number of points and neighborhood radius by layer were taken from the PointNet++ authors for outdoor and dense environments using MSG passing.

The second baseline is based on the KPConv architecture. We selected the KP-FCNN architecture provided by the authors for outdoor scenes [30]. It is composed of a five-layer network, where each layer contains two convolutional blocks, as originally proposed by the ResNet authors [32]. We used $dl_0 = 6$ cm, inspired by the value used by the authors for the Semantic3D dataset.

5.2.2. Implementation Details

We fixed similar training parameters between both baselines (Pointnet++ [29] and KPConv [30]) in order to compare their performance. As pre-processing, point clouds are sub-sampled on a grid, keeping one point per voxel (voxel size of 6 cm). Models learn, validate, and test with these data. Then, when testing, we perform inference with the under-sampled point clouds and then give the labels in "full resolution" with a KNN of the probabilities (not the labels). Spheres were computed in pre-processing (before the training stage) in order to reduce the computational cost. During training, we selected the spheres by class (class of the center point of the sphere) so that the network considered all the classes at each epoch, which greatly reduces the problem of class imbalance of the dataset. At each epoch, we took one point cloud from the dataset ($T_i$ for CARLA and $S_i$ for Paris) and set the number of spheres seen in this point cloud to 100.

Two features were included as input: RGB color information and height of points ($z$). In order to prevent overfitting, we included geometric data augmentation techniques: elastic distortion, random Gaussian noise with $\sigma = 0.1$ m and clip at 0.05 m, random rotations around $z$, anisotropic random scale between 0.8 and 1.2, and random symmetry around the $x$ and $y$ axes. We included the following transformations to prevent overfitting due to color information: chromatic jitter with $\sigma = 0.05$, and random dropout of RGB features with a probability of 20%.

For training, we selected the loss function as the sum of Cross Entropy and Power Jaccard with $p = 2$ [33]. We used a patience of 50 epochs (no progress in the validation set) and the optimizer ADAM with a default learning rate of 0.001. Both experiments were implemented using the Torch Points3D library [34] using a GPU NVIDIA Titan X with 12Go RAM.

Parameters presented in this section were chosen from a set of experiments varying the loss function (Cross Entropy, Focal Loss, Jaccard, and Power Jaccard) and input features (RGB and $z$ coordinate or only $z$ coordinate or only RGB or only 1 as input feature). The best results were obtained with the reported parameters.

5.2.3. Quantitative Results

Prediction of test point clouds was performed by using a sphere-based approach using a regular grid and maximum voting scheme. In this case, the spheres' centers were calculated to keep the intersections of spheres at 1/3 of their radius ($r = 10$ m as done during the training).

We report the obtained results in Table 2. We obtained an overall 13.9 % mIoU for Point-Net++ and 37.5 % mIoU for KPConv. This remains low for state-of-the-art architectures. This shows the difficulty and the wide variety of classes present in this Paris-CARLA-3D dataset. We can also see poorer results on synthetic data due to the greater variety of objects between CARLA cities, while, for the real data, the test data are very close to the training data.

**Table 2.** Results in semantic segmentation task using PointNet++ and KPConv architectures on our dataset, Paris-CARLA-3D. Results are mIoU in %. For $S_0$ and $S_3$, training set is $S_1$, $S_2$. For $T_1$ and $T_7$, training set is $T_2$, $T_3$, $T_4$ and $T_5$. Overall mIoU is the mean IoU on the whole test sets (real and synthetic).

| Model | Paris | | CARLA | | Overall |
|---|---|---|---|---|---|
| | $S_0$ | $S_3$ | $T_1$ | $T_7$ | mIoU |
| PointNet++ [29] | 13.9 | 25.8 | 4.0 | 12.0 | 13.9 |
| KPConv [30] | 45.2 | 62.9 | 16.7 | 25.3 | **37.5** |

5.2.4. Qualitative Results

Semantic segmentation of point clouds is better on Paris than on CARLA in all evaluated scenarios. This is an expected behavior because class variability and scene configurations are much more complex in the synthetic dataset. By way of an example, Figures 3–6 display the predicted labels and ground truth from the test sets of Paris and CARLA data. These images were obtained from the KPConv architecture.

From the qualitative results of semantic segmentation, it is evidenced the complexity of our proposed dataset. In the case of Paris data, color information is discriminant enough to separate sidewalks, roads, and road-lines. This is an expected behavior because the point clouds were from the same town and were acquired the same day. However, in CARLA point clouds, color information in ground-like classes changed between different towns. Additionally, in some towns, such as $T_2$, we included rain during simulations, visible in the color of the road. This characteristic makes the learning stage even more difficult.

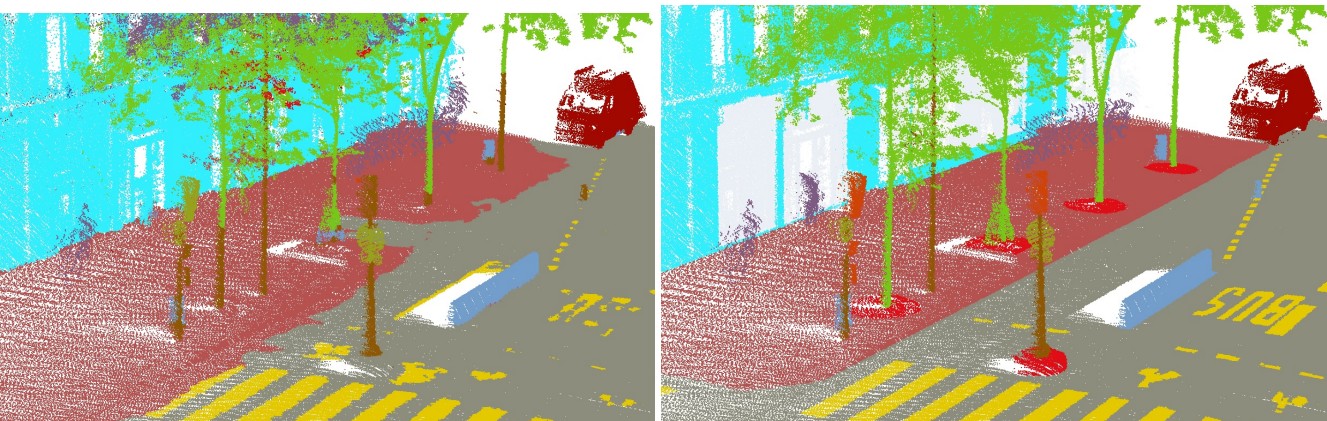

**Figure 3. Left**, prediction in $S_0$ test set of Paris data using KPConv model. **Right**, ground truth.

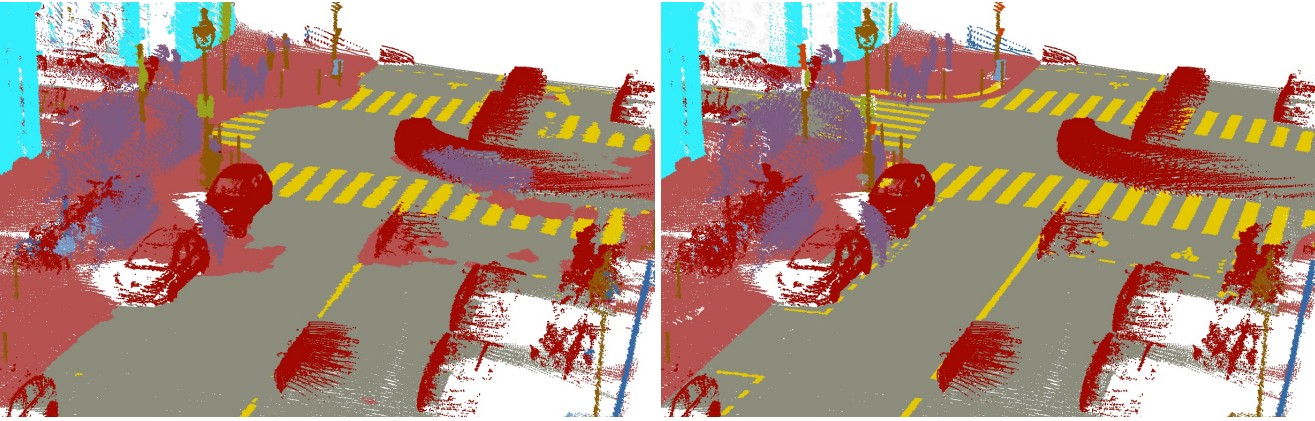

**Figure 4. Left**, prediction in $S_3$ test set of Paris data using KPConv model. **Right**, ground truth.

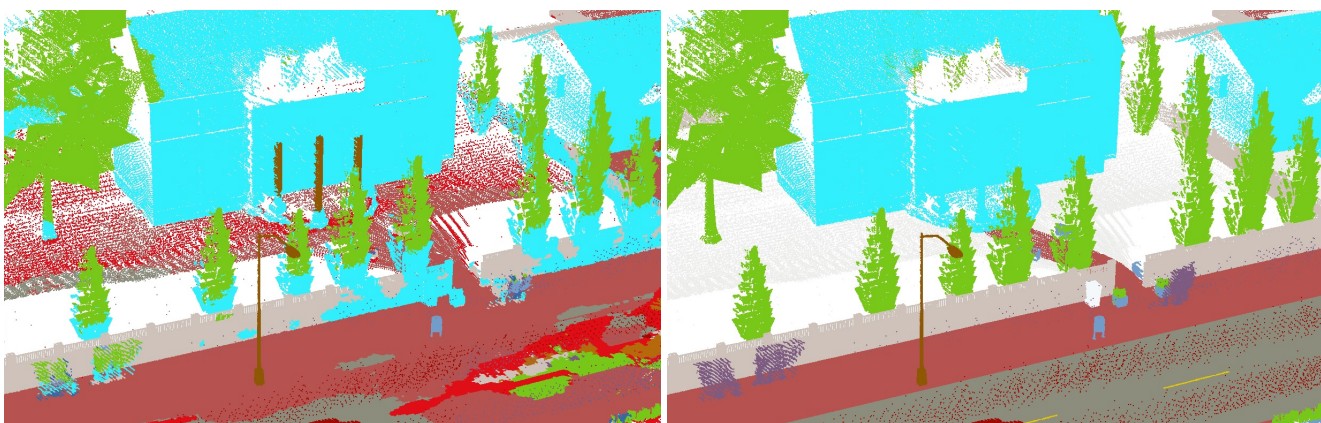

**Figure 5. Left**, prediction in $T_1$ test set of CARLA data using KPConv model. **Right**, ground truth.

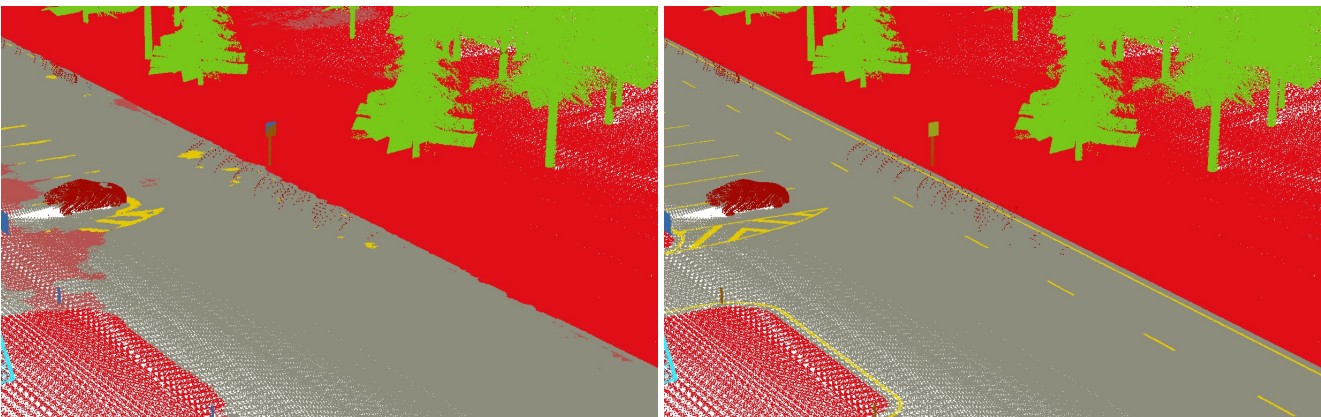

**Figure 6. Left**, prediction in $T_7$ test set of CARLA data using KPConv model. **Right**, ground truth.

### 5.2.5. Influence of Color

We studied the influence of color information during training in the PC3D dataset. In Table 3, we report the obtained results on the test set of semantic segmentation using the KPConv architecture without RGB features. The rest of the training parameters were the same as in the previous experiment. We can see that even if the colorization of the point cloud can create artifacts during the projection step (from the difference in point of view between the LiDAR sensor and the cameras or from the presence of moving objects), the use of the color modality in addition to geometry clearly improved the segmentation results.

**Table 3.** Results in semantic segmentation task using KPConv [30] architecture on our PC3D dataset with and without RGB colors on LiDAR points. Results are mIoU in %. For $S_0$ and $S_3$, training set is $S_1$, $S_2$. For $T_1$ and $T_7$, training set is $T_2$, $T_3$, $T_4$ and $T_5$. Overall mIoU is the mean IoU on the whole test sets (real and synthetic).

| Model | Paris | | CARLA | | Overall |
|---|---|---|---|---|---|
| | $S_0$ | $S_3$ | $T_1$ | $T_7$ | mIoU |
| KPConv w/o color | 39.4 | 41.5 | 35.3 | 17.0 | 33.3 |
| KPConv with color | 45.2 | 62.9 | 16.7 | 25.3 | **37.5** |

### 5.2.6. Transfer Learning

Transfer learning (TL) was performed with the aim of demonstrating the use of synthetic point clouds generated by CARLA to perform semantic segmentation on real-world point clouds. We selected the model with the best performance in the point clouds of the test set from CARLA data, i.e., taking the KPConv architecture (pre-trained on urban

towns $T_2$, $T_3$, and $T_5$ since real data are urban data). Then, we took it as a pre-training stage with Paris data.

We carried out different types of experiments as follows: (1) Predict test point clouds of Paris data using the best model obtained in urban towns from CARLA without training in Paris data (*no fine-tuning*); (2) Freeze the whole model except the last layer; (3) Freeze the feature extractor of the network; (4) No frozen parameters; (5) Training a model from scratch using only Paris training data. These scenarios were selected to evaluate the relevance of learned features in CARLA and their capacity to discriminate classes in Paris data. Results are presented in Table 4. The best results using TL were obtained in scenario 4: the model pre-trained in CARLA without frozen parameters during fine-tuning on Paris data. However, scenario 5 (i.e., *no transfer*) ultimately showed superior results.

**Table 4.** Results in transfer learning for the semantic segmentation task using KPConv architecture on our PC3D dataset. Results are mIoU in %. Pre-training was done using urban towns from CARLA ($T_2$, $T_3$, and $T_5$). *No fine-tuning*: the model was pre-trained on CARLA data without fine-tuning on Paris data. *No frozen parameters*: the model was pre-trained on CARLA without frozen parameters during fine-tuning on Paris data. *No transfer*: the model was trained only on the Paris training set.

| Transfer Learning Scenarios | Paris | | Overall |
|:---:|:---:|:---:|:---:|
| | $S_0$ | $S_3$ | mIoU |
| *No fine-tuning* | 20.6 | 17.7 | 19.2 |
| *Freeze except last layer* | 24.1 | 31.0 | 27.6 |
| *Freeze feature extractor* | 29.0 | 41.3 | 35.2 |
| *No frozen parameters* | 42.8 | 50.0 | 46.4 |
| *No transfer* | 45.2 | 62.9 | **51.7** |

From Table 4, a first finding is that the current model trained on synthetic data cannot be directly applied to real-world data (the *no fine-tuning* row). This is an expected result, because objects and class distributions in CARLA towns are different from real-world ones.

We may also observe that the performance of *no frozen parameters* is lower than that of *no transfer*: pre-training the network on the synthetic and fine-tuning on the real data decreases the performance compared to training directly on the real dataset. Alternatives are now introduced in order to close the existing gap between synthetic and real data, such as domain adaptation methods.

## 6. Instance Segmentation (IS) Task

The ability to detect instances in dense point clouds of outdoor environments can be useful for cities for urban space management (for example, to have an estimate of the occupancy of parking spaces through fast mobile mapping) or for building the prior map layer for HD maps in autonomous driving.

We provide instance annotations as follows: in Paris data, instances of *vehicle* class were manually point-wise annotated; in CARLA data, *vehicle* and *pedestrian* instances were automatically obtained by the CARLA simulator. Figure 7 illustrates the instance annotation of vehicles in $S_3$ Paris data. We found that pedestrians in Paris data were too close to each other to be recognized as separate instances (Figure 8).

### 6.1. Task Protocol

We introduce the task protocol to evaluate the instance segmentation methods in our dataset.

Evaluation of the performance in the instance segmentation task is different to that in the semantic segmentation task. Inspired by [35] on *things*, we report Segment Matching (SM) and Panoptic Quality (PQ), with $IoU = 0.5$ as the threshold to determine well-

predicted instances. We also report the mean IoU, based on IoU by instance $i$ ($IoU_i$), calculated as follows:

$$IoU_i = \begin{cases} IoU & IoU \geq 0.5 \\ 0, & \text{otherwise} \end{cases} \tag{2}$$

A common issue in LiDAR scanning is the presence of far objects that are unrecognizable due to the small number of points. In the semantic segmentation task, such objects do not affect evaluation metrics, due to their low rate. However, in the instance segmentation task, they may considerably affect the evaluation of the algorithms. In order to provide an evaluation metric having relevance, metrics are computed only with instances closer than $d = 20\,\text{m}$ to the mobile system.

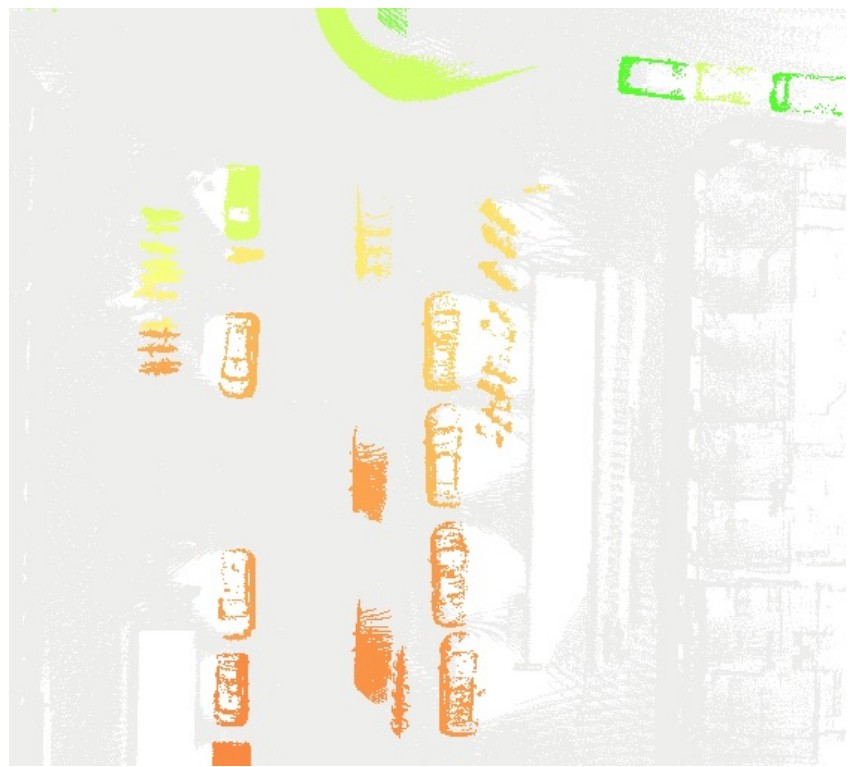

**Figure 7.** Instances of vehicles in $S_3$ test set (Paris data).

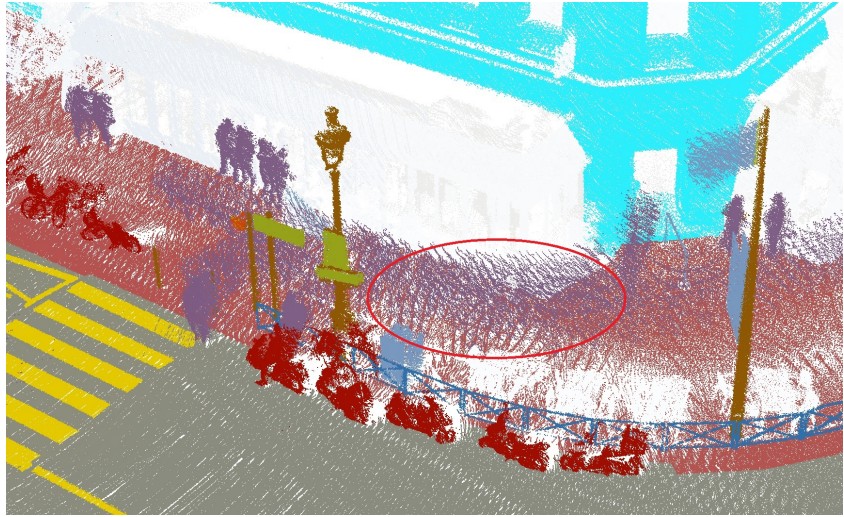

**Figure 8.** Pedestrians in Paris data: we can see inside the red circle the difficulty of differentiating the instances of pedestrians.

*6.2. Experiments: Setting a Baseline*

In this section, we present a baseline for the instance segmentation task and its evaluation with the introduced metrics. We propose a hybrid approach, combining deep learning and mathematical morphology, to predict instance labels. We report the obtained results for each point cloud of the test sets.

As presented by [36], urban objects can be classified using geometrical and contextual features. In our case, we start from already predicted *things* classes (vehicles and pedestrians, in this case) with the best model introduced in Section 5.2.3, i.e., using the KPConv architecture. Then, instances are detected by using Bird's Eye View (BEV) projections and mathematical morphology.

We computed the following BEV projections (with a pixel resolution of 10 cm) for each class:

- Occupancy image ($I_b$)—binary image with presence or not of *things* class;
- Elevation image ($I_h$)—stores the maximal elevation among all projected points on the same pixel;
- Accumulation image ($I_{acc}$)—stores the number of points projected on the same pixel.

At this point, three BEV projections were computed for each class: occupancy ($I_b$), elevation ($I_h$), and accumulation ($I_{acc}$). In the following sections, we describe the proposed algorithms to separate the vehicle and pedestrian instances. We highlight that these methods rely on the labels predicted in the semantic segmentation task (Section 5.2.1) using the KPConv architecture.

### 6.2.1. Vehicles in Paris and CARLA Data

One of the main challenges of this class is the high variability due to the different types of objects that it contains: cars, motorbikes, bikes, and scooters. Additionally, it also includes moving and parked vehicles, which makes it challenging to determine object boundaries.

From BEV projections, vehicle detection is performed as follows:

1. Discard the predicted points of the vehicle if the *z* coordinate is greater than 4 m in $I_h$;
2. Connect close components with two consecutive morphological dilations of $I_b$ by a square of 3-pixel size;
3. Fill holes smaller than ten pixels inside each connected component; this is performed with a morphological area closing;
4. Discard instances with less than 500 points in $I_{acc}$;
5. Discard instances not surrounded by ground-like classes in $I_b$.

### 6.2.2. Pedestrians in CARLA Data

As mentioned earlier for vehicles, the pedestrian class may contain moving objects. This implies that object boundaries are not always well-defined.

We followed a similar approach as described previously for vehicle instances based on the semantic segmentation results and BEV projections. We first discarded pedestrian points if the *z* coordinate was greater than 3 m in $I_h$, and then connected close components and filled small holes, as described for the vehicle class; we then discarded instances with less than 100 points in $I_{acc}$ and, finally, discarded instances not surrounded by ground-like classes in $I_b$.

### 6.2.3. Quantitative Results

For vehicles and pedestrians, instance labels of BEV images were back-projected to 3D data in order to provide point-wise predictions. In Table 5, we report the obtained results in instance segmentation using the proposed approach. These results are the first of a method allowing instance segmentation on dense points clouds from 3D mapping, and we hope that it will inspire future methods.

**Table 5.** Results on test sets of Paris-CARLA-3D for the instance segmentation task. SM: Segment Matching. PQ: Panoptic Quality. *mIoU*: mean IoU. All results are in %.

|  | # Instances | SM | PQ | mIoU |
|---|---|---|---|---|
| $S_0$—Vehicles | 10 | 90.0 | 70.9 | 81.6 |
| $S_3$—Vehicles | 86 | 32.6 | 40.5 | 28.0 |
| $T_1$—Vehicles | 41 | 17.1 | 20.4 | 14.2 |
| $T_7$—Vehicles | 27 | 74.1 | 72.6 | 61.2 |
| $T_1$—Pedestrians | 49 | 18.4 | 17.0 | 13.9 |
| $T_7$—Pedestrians | 3 | 100.0 | 9.0 | 66.0 |
| **Mean** | 216 | 55.3 | 38.4 | 44.2 |

### 6.2.4. Qualitative Results

In our proposed baseline, instances are separated using BEV projections and geometrical features based on semantic segmentation labels. In some cases, as presented in Figure 9, 2D projections can merge objects in the same instance label if they are too close.

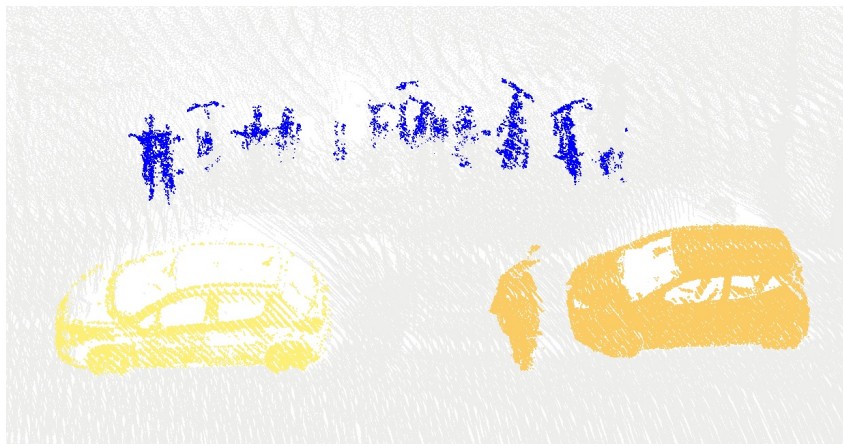

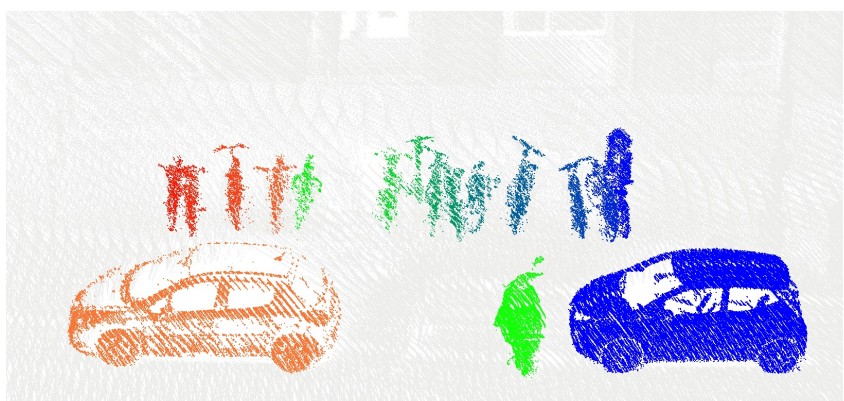

**Figure 9. Top**, vehicle instances from our proposed baseline using BEV projections and geometrical features in $S_3$ Paris data. **Bottom**, ground truth.

Close objects and instance intersections are challenging for the instance segmentation task. The former can be tackled by using approaches based directly on 3D data. For the latter, we provide timestamp information by point in each PLY file. The availability of this feature may be useful for future approaches.

Semantic segmentation and instance segmentation could be unified in one task, Panoptic Segmentation (PS): this is a task that has recently emerged in the context of scene understanding [35]. We leave this for future works.

## 7. Scene Completion (SC) Task

The scene completion (SC) task consists of predicting the missing parts of a scene (which can be in the form of a depth image, a point cloud, or a mesh). This is an important problem in 3D mapping due to holes from occlusions and holes after the removal of unwanted objects, such as vehicles or pedestrians (see Figure 10). It can be solved in the form of 3D reconstruction [37], scan completion [38], or, more specifically, methods to fill holes in a 3D model [39].

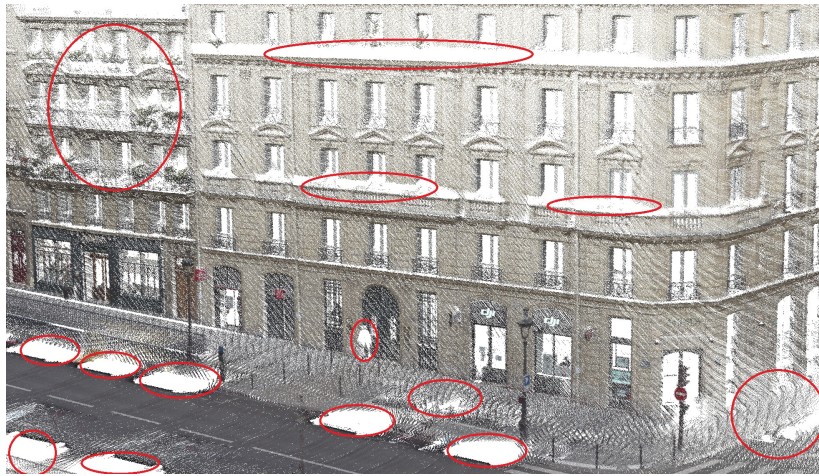

**Figure 10.** Paris data after removal of vehicles and pedestrians. Zones in red circles show the interest in conducting scene completion for 3D mapping, in order to fill holes from removed pedestrians, parked cars, and from the occlusion of other objects, and also to improve the sampling of points in areas far from the LiDAR.

Semantic scene completion (SSC) is the task of filling the geometry as well as predicting the semantics of the points, with the aim that the two tasks carried out simultaneously benefit each other (survey of SSC in [40]). It is also possible to jointly predict the geometry and color during scene completion, as in SPSG [41]. For now, we only evaluate the geometry prediction, as we leave the prediction of simultaneous geometry, semantics, and color for future work.

The vast majority of the existing methods of scene completion (SC) work focus on small indoor scenes, while, in our case, we have a dense outdoor environment with our Paris-CARLA-3D dataset. Completing outdoor LiDAR point clouds is more challenging than data obtained from RGB-D images acquired in indoor environments, due to the sparsity of points obtained using LiDAR sensors. Moreover, larger occluded areas are present in outdoor scenes, caused by static and temporary foreground objects, such as trees, parked vehicles, bus stops, and benches. SemanticKITTI [7] is a dataset conducting scene completion (SC) and semantic scene completion (SSC) on LiDAR data, but they use only one single scan as input, with a target (ground truth) being the accumulation of all LiDAR scans. In our dataset, we seek to complete the "holes" from the accumulation of all LiDAR scans.

### 7.1. Task Protocol

We introduce the task protocol to perform scene completion on PC3D. Our goal is to predict a more complete point cloud. First, we extract random small chunks from the original point cloud that we transform into a discretized regular 3D grid representation containing the Truncated Signed Distance Function (TSDF) values, which expresses the distance from each voxel to the surface represented by the point cloud. Then, we use a neural network to predict a new TSDF and finally, we extract a point cloud from that TSDF that should be more complete that the input. We used as TSDF the classical signed point-to-plane distance to the closest point of the point cloud as in [42]. Our original point

cloud is already incomplete due to the occlusions caused by static objects and the sparsity of the scans. To overcome this incompleteness, we make the point cloud more incomplete by removing 90% of the points (by *scan_index*), and use the incomplete data to compute the TSDF input of the neural network. Moreover, we use the original point cloud containing all of the points as the ground truth and compute the target TSDF. Our approach is inspired by the work done by SG-NN [43] and we do this in order to learn to complete the scene in a self-supervised way. Removing points according to their *scan_index* allows us to create larger "holes" than by removing points at random. For the chunks, we used a grid size of $128 \times 128 \times 128$ and a voxel size of 5 cm (compared to the voxel size of 2 cm used for indoor scenes in SG-NN [43]). Dynamic objects, pedestrians, vehicles, and unlabeled points are first removed from the data using the ground truth semantic information.

To evaluate the completed scene, we use the Chamfer Distance (CD) between the original $P_1$ and predicted $P_2$ point clouds:

$$CD = \frac{1}{|P_1|} \sum_{x \in P_1} \min_{y \in P_2} ||x - y||_2 + \frac{1}{|P_2|} \sum_{y \in P_2} \min_{x \in P_1} ||y - x||_2 \tag{3}$$

In a self-supervised context, not having the ground truth and having the predicted point cloud more complete than the target places some limitations on using the CD metric. For this, we introduce a mask that needs to be used to compute the CD only on the points that were originally available. The mask is simply a binary occupancy grid on the original point cloud.

We extract the random chunks as explained previously for Paris (1000 chunks per point cloud) and CARLA (3000 chunks per town) and provide them along with the dataset for future research on scene completion.

### 7.2. Experiments: Setting a Baseline

In this section, we present a baseline for scene completion using the SG-NN network [43] to predict the missing points (SG-NN predicts only the geometry and not the semantics nor the color). In SG-NN, they use volumetric fusion [44] to compute a TSDF from range images, which cannot be used on LiDAR point clouds. For this, we compute a different TSDF from the point clouds.

Using the cropped chunks, we estimate the normal at each point using PCA as in [42] with $n = 30$ neighbors and obtain a consistent orientation using the LiDAR sensor position provided with the points. Using the normal information, we use the SDF introduced in [42], due to its simplicity and the ease of vectorizing, which reduces the data generation complexity. After obtaining the SDF volumetric representation, we convert the values to voxel units and truncate the function at three voxels, which results in a 3D sparse TSDF volumetric representation that is similar to the input of SG-NN [43]. For the target, we use all the points available in the original point cloud, and for the input, we keep 10% of points (by the scan indices) in each chunk, in order to obtain the "incomplete" point cloud representation.

The resulting sparse tensors are then used for training and the network is trained for 20 epochs with ADAM and a learning rate of 0.001. The loss is a combination of Binary Cross Entropy (BCE) on occupancy and L1 Loss on TSDF prediction. The training was carried out on a GPU NVIDIA RTX 2070 SUPER with 8Go RAM.

In order to increase the number of samples and prevent overfitting, we perform data augmentation on the extracted chunks: random rotation around $z$, random scaling between 0.8 and 1.2, and local noise addition with $\sigma = 0.05$.

Finally, we extract a point cloud from the TSDF predicted by the network following an approach that is similar to the marching cubes algorithm [45], where we interpolate 1 point per voxel. Finally, we compute the CD (see Equation (3)) between the point cloud extracted from the predicted TSDF and the original point cloud (without dynamic objects) and use the introduced mask to limit the CD computation to known regions (voxels where we have points in the original point cloud).

### 7.2.1. Quantitative Results

Table 6 shows the results of our experiment on Paris-CARLA-3D data. We can see that the network makes it possible to create point clouds whose distance to the original cloud is clearly smaller.

For further metric evaluation, we provide the mean IoU and $\ell_1$ distance between the target and predicted TSDF values on the 2000 and 6000 chunks for Paris and CARLA test sets, respectively. The results are also reported in Table 6.

**Table 6.** Scene completion results on Paris-CARLA-3D. CD is the mean Chamfer Distance over 2000 chunks for the Paris test set ($S_0$ and $S_3$) and 6000 chunks for the CARLA test set ($T_1$ and $T_7$). $\ell_1$ is the mean $\ell_1$ distance between predicted and target TSDF measured in voxel units for 5 cm voxels and $mIoU$ is the mean Intersection over Union of TSDF occupancy. Both metrics are computed on known voxel areas. *ori* means original point cloud, *in* is input point cloud (10% of the original), *pred* is the predicted point cloud (computed from predicted TSDF).

| Test Set | $CD_{in \leftrightarrow ori}$ | $CD_{pred \leftrightarrow ori}$ | $\ell_{1\,pred \leftrightarrow tar}$ | $mIoU_{pred \leftrightarrow tar}$ |
|:---:|:---:|:---:|:---:|:---:|
| $S_0$ and $S_3$ (Paris) | 16.6 cm | 10.7 cm | 0.40 | 85.3% |
| $T_1$ and $T_7$ (CARLA) | 13.3 cm | 10.2 cm | 0.49 | 80.3% |

### 7.2.2. Qualitative Results

Figure 11 shows the scene completion result on one point cloud chunk from the CARLA $T_1$ test set. Figure 12 shows the scene completion result on one chunk from the Paris $S_0$ test set. We can see that the network manages to produce point clouds quite close visually to the original, despite having as input a sparse point cloud with only 10% of the points of the original.

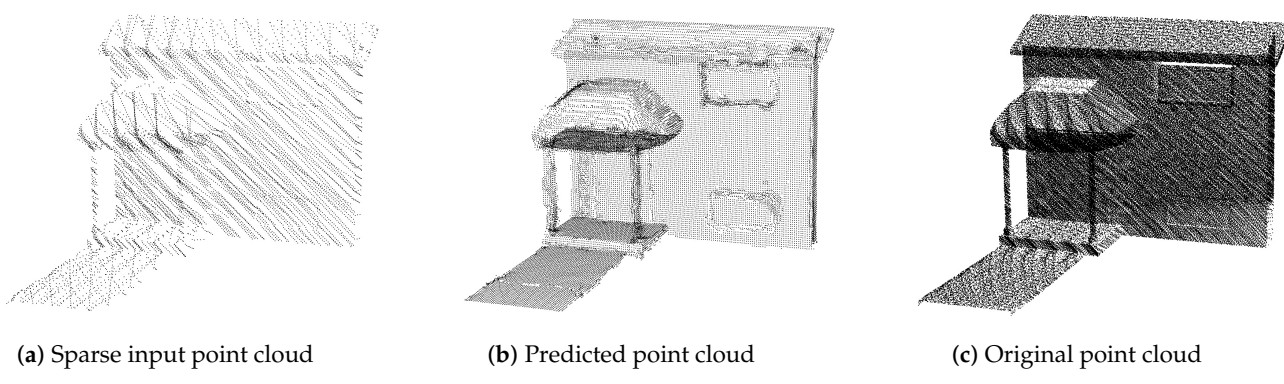

(**a**) Sparse input point cloud          (**b**) Predicted point cloud          (**c**) Original point cloud

**Figure 11.** Scene completion task for one chunk point cloud in Town1 ($T_1$) of CARLA test data (training on CARLA data).

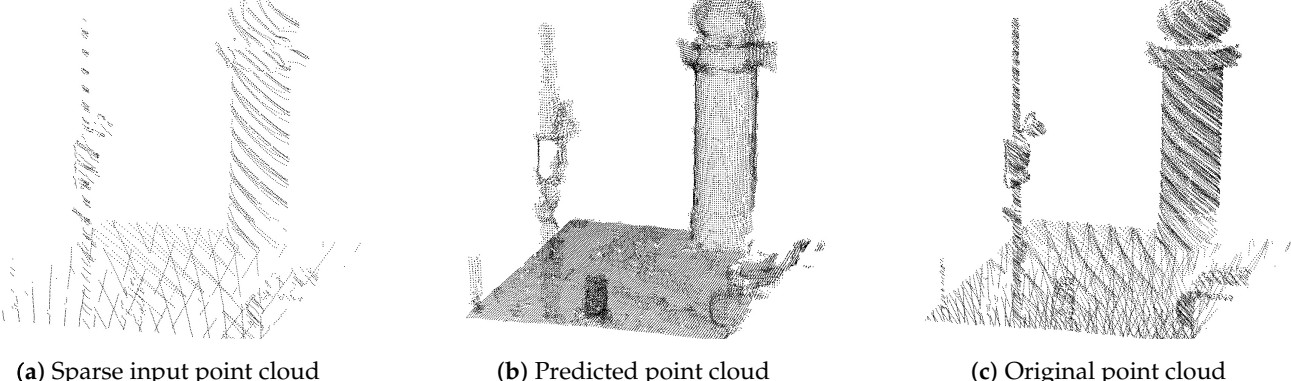

(**a**) Sparse input point cloud          (**b**) Predicted point cloud          (**c**) Original point cloud

**Figure 12.** Scene completion task for one chunk point cloud in Soufflot0 ($S_0$) of Paris test data (training on Paris data).

### 7.2.3. Transfer Learning with Scene Completion

Using both synthetic and real data of Paris-CARLA-3D, we tested the training of a scene completion model on CARLA synthetic data to test it on Paris data. With the objective of scene completion on real data chunks (Paris $S_0$ and $S_3$), we tested three training scenarios: (1) Training only on real data with Paris training set; (2) Training only on synthetic data with CARLA training set; (3) Pre-train on synthetic data then fine-tune on real data. The results are shown in Table 7. We can see that the Chamfer Distance (CD) is better for the model trained only on synthetic CARLA data: the network is attempting to fill a local plane in large missing regions and smoothing the rest of the geometry. This is an expected behavior, because of the handcrafted geometry present in CARLA, where planar geometric features are predominantly present. Point clouds of real outdoor scenes are not easily obtained and the need to complete missing geometry is becoming increasingly important in vision-related tasks; here, we can see the value of leveraging the large amount of synthetic data present in CARLA to pre-train the network and fine-tune it on other smaller datasets such as Paris when not enough data are available. As we can see in Table 7, pre-training on CARLA and then fine-tuning on Paris allows us to obtain the best predicted TSDF ($\ell_1$ and *mIoU*) and point cloud (Chamfer Distance).

**Table 7.** Results of transfer learning for the scene completion task. CD is the mean Chamfer Distance between point clouds. $\ell_1$ is the mean $\ell_1$ distance between predicted and target TSDF measured in voxel units for 5 cm voxels and *mIoU* is the mean Intersection over Union of TSDF occupancy. The mean is over 2000 chunks for Paris data. *ori* means original point cloud, *in* is input point cloud (10 % of the original), *pred* for CD is the predicted point cloud (computed from predicted TSDF), *pred* for IoU, $\ell_1$ is the predicted TSDF, and *tar* is the target TSDF.

| Test Set: $S_0$ and $S_3$ Paris data | $CD_{in \leftrightarrow ori}$ | $CD_{pred \leftrightarrow ori}$ | $\ell_{1pred \leftrightarrow tar}$ | $mIoU_{pred \leftrightarrow tar}$ |
|---|---|---|---|---|
| Trained only on Paris | 16.6 cm | 10.7 cm | 0.40 | 85.3% |
| Trained only on CARLA | 16.6 cm | 8.0 cm | 0.48 | 84.0% |
| Pre-trained CARLA, fine-tuned on Paris | 16.6 cm | 7.5 cm | 0.35 | 88.7% |

## 8. Conclusions

We presented a new dataset called Paris-CARLA-3D. This dataset is made up of both synthetic data (700 M points) and real data (60 M points) from the same LiDAR and camera mobile platform. Based on this dataset, we presented three classical tasks in 3D computer vision (semantic segmentation, instance segmentation, and scene completion) with their evaluation protocol as well as a baseline, which will serve as starting points for future work using this dataset.

On semantic segmentation (the most common task in 3D vision), we tested two state-of-the-art methods, PointNet++ and KPConv, and showed that KPConv obtains the best results (37.5% overall mIoU). We also presented a first instance detection method on dense point clouds from mapping systems (with vehicle and pedestrian instances for synthetic data and vehicle instances for real data). For the scene completion task, we were able to adapt a method used for indoor data with RGB-D sensors to outdoor LiDAR data. Even with a simple formulation of the surface, the network manages to learn complex geometries, and, moreover, by using the synthetic data as pre-training, the method obtains better results on the real data.

**Author Contributions:** Methodology and writing J.-E.D.; data annotation, methodology and writing D.D. and J.P.R.; supervision and reviews S.V.-F., B.M. and F.G. All authors have read and agreed to the published version of the manuscript.

**Funding:** This research was partially funded by REPLICA FUI 24 project.

**Data Availability Statement:** The dataset is available at the following URL: https://npm3d.fr/paris-carla-3d, accessed on 15 October 2021.

**Conflicts of Interest:** The authors declare no conflict of interest.

## Appendix A. Complementary on Paris-CARLA-3D Dataset

*Appendix A.1. Class Statistics*

From CARLA data, as occurs in real-world scenarios, not every class is present in every town: eleven classes are present in all towns (*road, building, sidewalk, vegetation, vehicles, road-line, fence, pole, static, dynamic, traffic sign*), three classes in six towns (*unlabeled, wall, pedestrian*), three classes in five towns (*terrain, guard-rail, ground*), two classes in four towns (*bridge, other*), one class in three towns (*water*), and two classes in two towns (*traffic light, rail-track*).

In Paris data, class variability is smaller than in CARLA data. This is a desired (and expected) feature of these point clouds because they correspond to the same town. However, as is the case with CARLA towns, not every class is present in every point cloud: twelve classes are present in all point clouds (*road, building, sidewalk, road-line, vehicles, other, unlabeled, static, pole, dynamic, pedestrian, traffic sign*), three classes in five point clouds (*vegetation, fence, traffic light*), one class in two point clouds (*terrain*), and seven classes in any point cloud (*wall, sky, ground, bridge, rail-track, guard-rail, water*).

Table A1 shows the detailed statistics of the classes in the Paris-CARLA-3D dataset.

**Table A1.** Class distribution in Paris-CARLA-3D dataset (in %). Columns headed by $S_i$ are Soufflot from Paris data and $T_j$ are towns from CARLA data.

| Class | Paris | | | | | | CARLA | | | | | | |
|---|---|---|---|---|---|---|---|---|---|---|---|---|---|
| | $S_0$ | $S_1$ | $S_2$ | $S_3$ | $S_4$ | $S_5$ | $T_1$ | $T_2$ | $T_3$ | $T_4$ | $T_5$ | $T_6$ | $T_7$ |
| unlabeled | 0.9 | 1.5 | 3.9 | 3.2 | 1.9 | 0.9 | 5.8 | 2.9 | - | 7.6 | 0.0 | 6.4 | 1.8 |
| building | 14.9 | 18.9 | 34.2 | 36.6 | 33.1 | 32.9 | 6.8 | 22.6 | 15.3 | 4.5 | 16.1 | 2.6 | 3.3 |
| fence | 2.3 | 0.6 | 0.7 | 0.8 | - | 0.4 | 1.0 | 0.6 | 0.0 | 0.5 | 3.8 | 1.5 | 0.6 |
| other | 2.1 | 3.4 | 6.7 | 2.2 | 2.5 | 0.4 | - | - | - | - | 0.1 | 0.1 | 0.1 |
| pedestrian | 0.2 | 1.0 | 0.6 | 1.0 | 0.7 | 0.7 | 0.1 | 0.2 | 0.1 | 0.0 | - | 0.1 | 0.0 |
| pole | 0.6 | 0.9 | 0.6 | 0.8 | 0.7 | 1.1 | 0.6 | 0.6 | 4.2 | 0.8 | 0.8 | 0.4 | 0.3 |
| road-line | 3.8 | 3.7 | 2.4 | 4.1 | 3.5 | 3.4 | 0.2 | 0.2 | 2.9 | 1.6 | 2.2 | 1.3 | 1.7 |
| road | 41.0 | 49.7 | 35.0 | 37.6 | 40.6 | 27.5 | 47.8 | 37.2 | 53.1 | 52.8 | 44.7 | 58.0 | 42.8 |
| sidewalk | 10.1 | 4.2 | 7.3 | 6.7 | 11.9 | 29.4 | 22.5 | 17.5 | 10.3 | 1.7 | 10.5 | 3.1 | 0.4 |
| vegetation | 18.5 | 9.0 | 0.1 | 0.3 | 0.1 | - | 8.7 | 10.8 | 2.7 | 12.8 | 4.6 | 8.1 | 23.1 |
| vehicles | 1.3 | 1.8 | 6.5 | 6.5 | 3.3 | 1.6 | 1.7 | 3.1 | 0.9 | 0.5 | 3.1 | 4.2 | 0.9 |
| wall | - | - | - | - | - | - | 1.9 | 3.6 | 1.4 | 5.4 | 5.3 | 3.4 | - |
| traffic sign | 0.1 | 0.4 | 0.1 | 0.1 | 0.3 | 0.1 | - | 0.0 | - | 0.1 | 0.0 | 0.0 | 0.1 |
| sky | - | - | - | - | - | - | - | - | - | - | - | - | - |
| ground | - | - | - | - | - | - | - | 0.0 | 0.2 | 1.4 | 0.3 | 0.1 | - |
| bridge | - | - | - | - | - | - | 1.7 | - | - | 0.7 | 6.6 | - | - |
| rail-track | - | - | - | - | - | - | - | - | 7.6 | - | 0.5 | - | - |
| guard-rail | - | - | - | - | - | - | 0.0 | - | - | 4.3 | - | 1.2 | 0.5 |
| static | 2.6 | 2.3 | 0.3 | 0.1 | 0.7 | 1.5 | 0.1 | 0.1 | - | - | - | - | - |
| traffic light | 0.1 | 0.2 | 0.1 | 0.1 | 0.1 | - | 0.8 | 0.5 | 0.3 | 0.3 | 0.3 | - | - |
| dynamic | 0.3 | 1.6 | 1.5 | 0.2 | 0.7 | 0.0 | 0.1 | 0.1 | 0.1 | 0.3 | 0.1 | 0.1 | 0.1 |
| water | - | - | - | - | - | - | 0.4 | - | 0.0 | - | - | - | 0.6 |
| terrain | 1.4 | 0.8 | - | - | - | - | - | - | 0.9 | 4.8 | 1.1 | 9.6 | 23.8 |
| **# Points** | **60 M** | | | | | | **700 M** | | | | | | |

*Appendix A.2. Instances*

The number of instances in ground truth varies over the point clouds. In the test set from Paris data, Soufflot0 ($S_0$) has 10 vehicles while Soufflot3 ($S_3$) has 86. This large difference occurs due to the presence of parked motorbikes and bikes.

With respect to CARLA data, it was observed that in urban towns such as Town1 ($T_1$), vehicle and pedestrian instances are mainly moving objects. This implies that during simulations, instances can have intersections between them, making their separation challenging.

In the CARLA simulator, the instances of the objects are given by their IDs. If a vehicle/pedestrian is seen several times, the same instance_id is used at different places. This is a problem in the evaluation capacity of detecting correctly the instances. This is why we have divided the CARLA instances using the timestamp of points: separate instances based on a timestamp gap with a threshold of 10 s for vehicles and 5 s for pedestrians.

## Appendix B. Images of the Dataset

Figures A1–A6 show top-view images of the different point clouds of the Paris-CARLA-3D dataset.

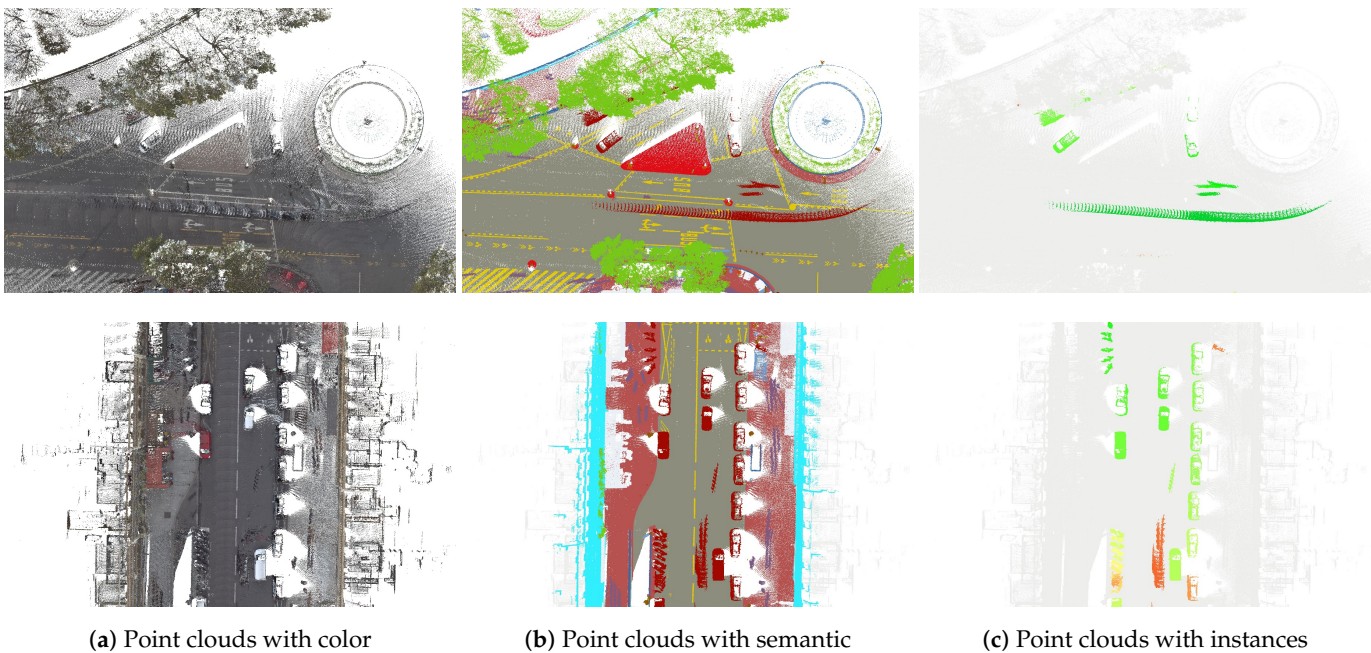

**(a)** Point clouds with color          **(b)** Point clouds with semantic          **(c)** Point clouds with instances

**Figure A1.** Paris training set. From **top** to **bottom**: $S_1$, $S_2$ (real data).

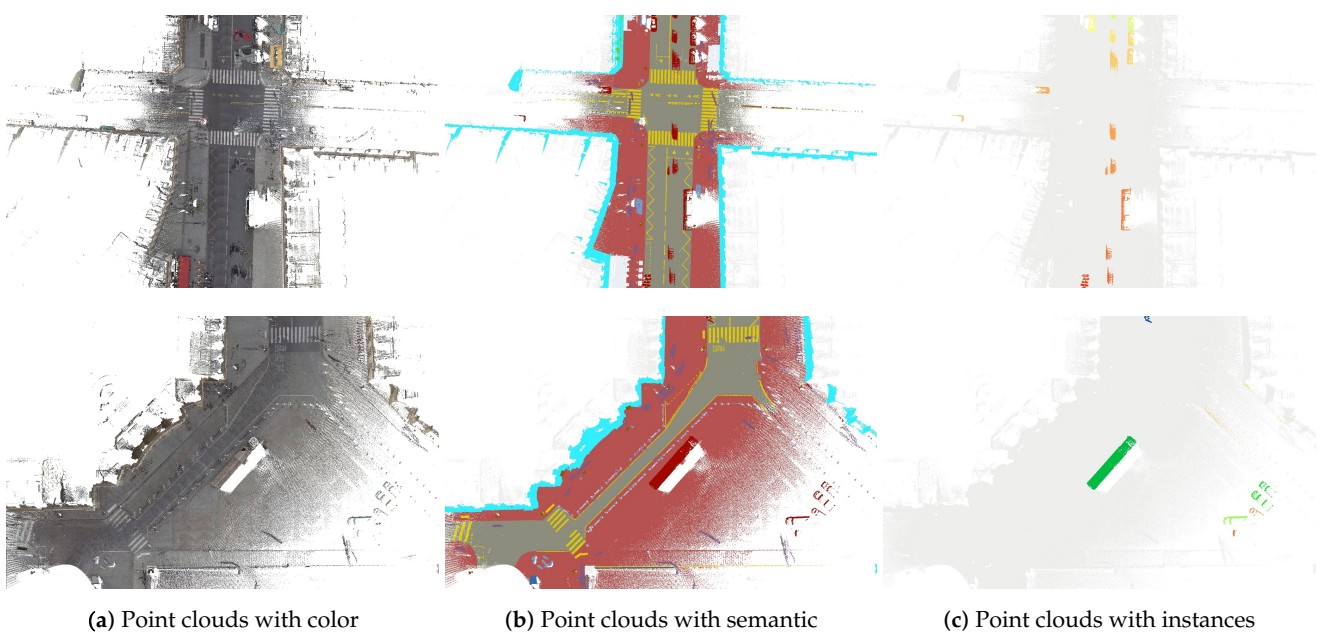

(**a**) Point clouds with color  (**b**) Point clouds with semantic  (**c**) Point clouds with instances

**Figure A2.** Paris validation set. From **top** to **bottom**: $S_4$, $S_5$ (real data).

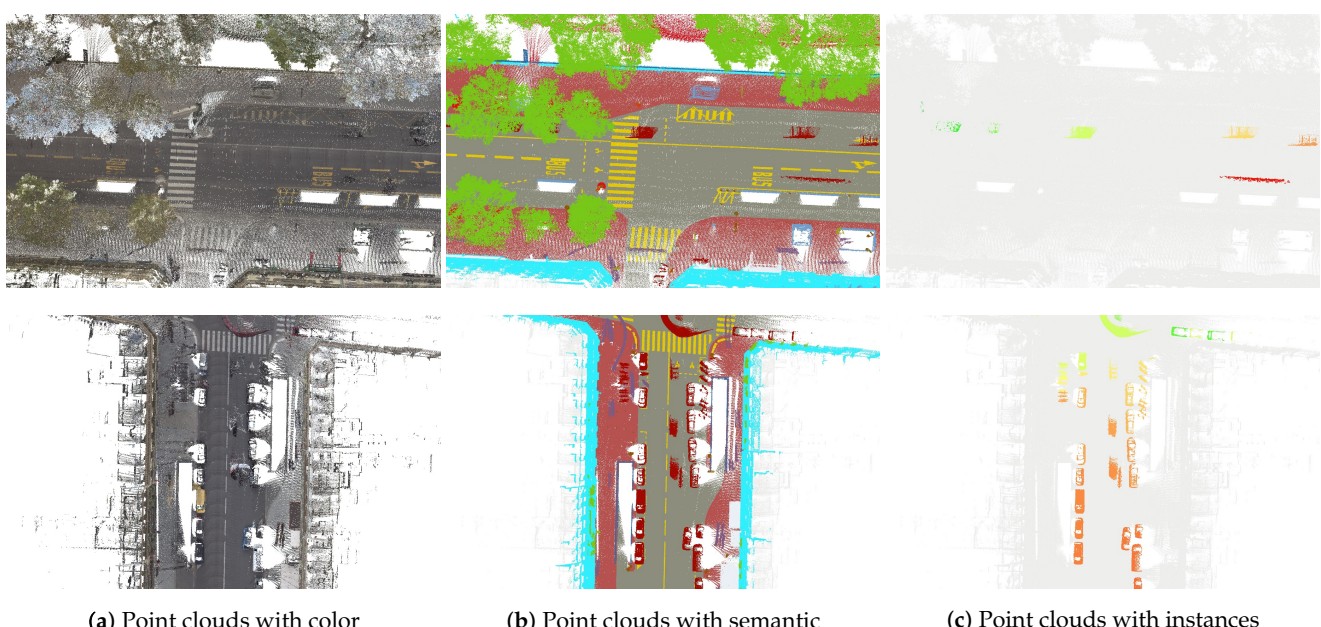

(**a**) Point clouds with color  (**b**) Point clouds with semantic  (**c**) Point clouds with instances

**Figure A3.** Paris test set. From **top** to **bottom**: $S_0$, $S_3$ (real data).

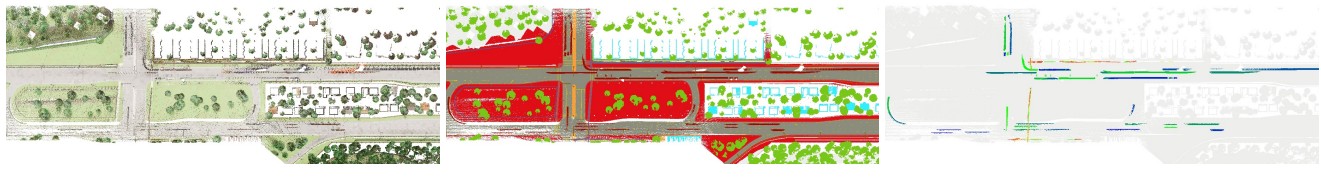

(**a**) Point clouds with color      (**b**) Point clouds with semantic      (**c**) Point clouds with instances

**Figure A4.** CARLA training set. From **top** to **bottom**: $T_2$, $T_3$, $T_4$, $T_5$ (synthetic data).

(**a**) Point cloud with color      (**b**) Point cloud with semantic      (**c**) Point cloud with instances

**Figure A5.** CARLA validation set. $T_6$ (synthetic data).

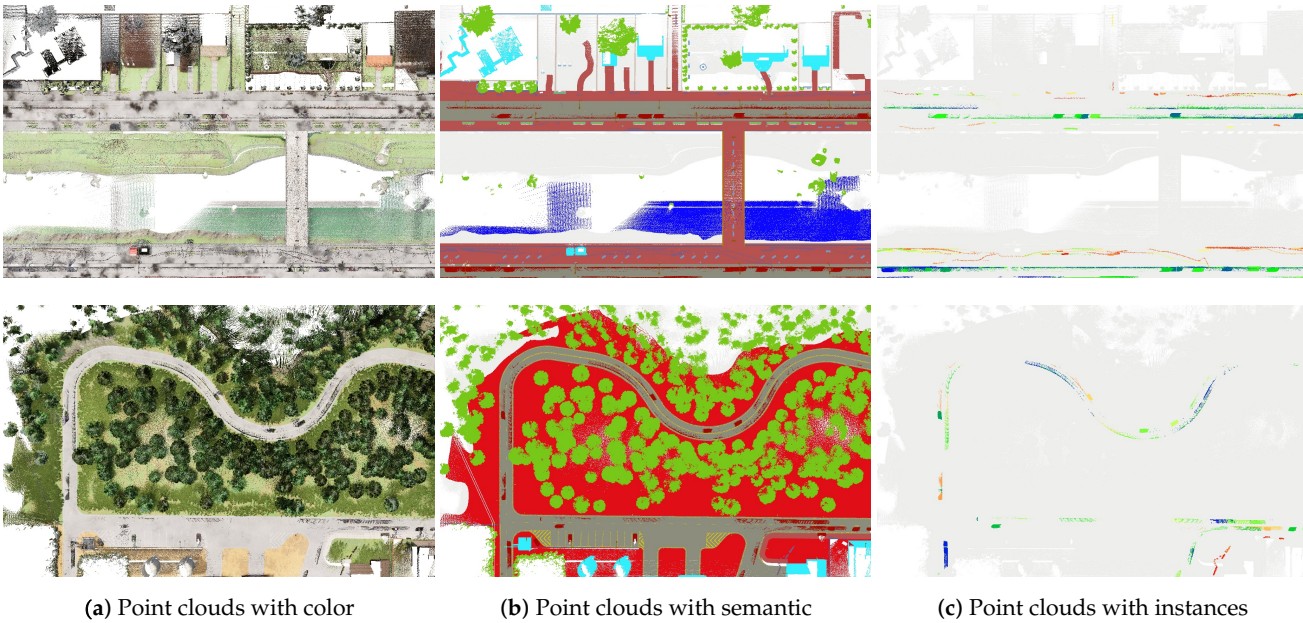

(**a**) Point clouds with color     (**b**) Point clouds with semantic     (**c**) Point clouds with instances

**Figure A6.** CARLA test set. From **top** to **bottom**: $T_1$, $T_7$ (synthetic data).

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
