# Peer review of "Paris-CARLA-3D: A Real and Synthetic Outdoor Point Cloud Dataset for Challenging Tasks in 3D Mapping"

_remotesensing, doi:10.3390/rs13224713_

Round 1

Reviewer 1 Report

The article consists of 21 pages, divided into 8 sections and contains
11 figures, 7 tables and 2 appendices (+ 1 table, + 6 figures). 43
references are cited.  

This work presents a dataset of several dense colored point clouds of
outdoor environments built by a mobile LiDAR and camera system,
entitled Paris-CARLA-3D (also denoted PC3D). In fact, the dataset is
composed of 2 sets of data: real data (acquired 4 in the city of
Paris) and synthetic data (generated with CARLA simulator). It is the
real platform that has been modeled and simulated in the open source
CARLA simulator. Moreover, a manual annotation of the classes using
the semantic tags of CARLA was made on the real data. The dataset is
yet not made publicly available but an url is provided.

The other contribution claimed by the authors the protocol and
experiments with baselines on three tasks (semantic segmentation, 32
instance segmentation, and scene completion) based on this dataset.

The article is well-written and easy to read. The quality of figures
is very good.

The synthesis of datasets based on 3D point clouds presented in Table
1 (Section 2. Related Datasets) is excellent. It allows the authors to
conclude that Paris-CARLA-3D is the only dataset to offer
annotations and protocols that allow for working on semantic,
instance, and scene completion tasks on dense point clouds for
outdoor mapping.

The construction of the dataset is clearly explained (Section 3) as
the dataset properties (Section 4).

However, Section 5.2 (Experiments: setting a baseline) should better
detailed, in particular the sentence: "We experimentally found that
selecting random spheres with a radius of r = 10m was a good
compromise between computational cost and performance."

Beyond the experiments carried out with the PointNet++ and KPConv
methods, the results of which are illustrated in Section
5.2.4. Qualitative results, the studies on the influence of color or
on transfer learning are very interesting and open up new perspectives
(works on domain adaptation).

* line 293 "We found that pedestrians in Paris data were too close to
each other to be recognized as separate instances." -> Please
illustrate this remark with a figure (as done for annotation of
vehicles in Fig. 7).

Once more, Section 6.2. Experiments: setting a baseline, the baseline
for the instance segmentation task should be more detailed (choice of
methods, etc.). Results and perspectives are relevant.

The third task described in Section 7 (Scene Completion) is again well
done and interesting results. As for the previous remarks, one can
sometimes regret that the descriptions are sometimes too short.

* Figure 10 show -> Figure 10 shows (line 443)
* Figures 11 show -> ?
* in the Table 7 -> in Table 7 (line 452)

The conclusion should better summarize the different contributions and
results.

Overall rating:
==========

A clear and interesting work that should be accepted with few minor
revisions even if the protocol of experiments should be described with
more details.

Reviewer 2 Report

This manuscript reports the Paris-CARLA-3D dataset, which is a benchmark of several dense colored point clouds of outdoor environments built by a mobile LiDAR and camera system. The presented dataset is of interest as the data is composed of two sets with synthetic data from the open-source CARLA simulator (700 million points) and real data acquired in the city of Paris (60 million points), hence the name Paris-CARLA-3D. I would like to accept this work and only have a few remarks.
1.    The use of synthetic data is the major highlight of your presented new dataset. However, the advantage of exploiting synthetic data is not clearly stated. It would be better to set a separated paragraph or section to elaborate why the use of synthetic data can bring us advantages (for example, can get denser and more evenly distributed points; reduce manual workload).
2.    You regard the protocol and experiments with baselines in three various tasks as a contribution to this work. I would say it is always nice to give some baseline results on the presented benchmark. However, it seems that for the semantic segmentation task, you only tested KPConv and PointNet++, which are not the state-of-the-art method anymore. I would suggest testing more recent methods on your proposed datasets. I would suggest trying at least one new method like RandLA-Net, the codes of which are available.
3.    It seems that there are still some other MLS datasets that are not mentioned, which are also opened to the public:
Li, X., Li, C., Tong, Z., Lim, A., Yuan, J., Wu, Y., Tang, J. and Huang, R., 2020, October. Campus3d: A photogrammetry point cloud benchmark for hierarchical understanding of outdoor scene. In Proceedings of the 28th ACM International Conference on Multimedia (pp. 238-246).
Zhu, J., Gehrung, J., Huang, R., Borgmann, B., Sun, Z., Hoegner, L., Hebel, M., Xu, Y. and Stilla, U., 2020. TUM-MLS-2016: An annotated mobile LiDAR dataset of the TUM city campus for semantic point cloud interpretation in urban areas. Remote Sensing, 12(11), p.1875.
Hu, Q., Yang, B., Khalid, S., Xiao, W., Trigoni, N. and Markham, A., 2021. Towards semantic segmentation of urban-scale 3d point clouds: A dataset, benchmarks and challenges. In Proceedings of the IEEE/CVF Conference on Computer Vision and Pattern Recognition (pp. 4977-4987).
I recommend mentioning and discussing these datasets in your manuscript as well.
4.    For scene completion, you implemented the filling of holes and use the scan as the input. Here, you use the Chamfer Distance (CD) between the original P1 and predicted P2 point clouds as the criterion for evaluation. However, it is not clear, how the ground truth is generated? Did you use the scan with filled holes (by removal of vehicles and pedestrians) as the ground truth? Or did you just measure the CD between the input and output dataset for evaluation? Moreover, in this task, which role the synthetic data play?
